# Identification of CAMTA Gene Family in *Heimia myrtifolia* and Expression Analysis under Drought Stress

**DOI:** 10.3390/plants11223031

**Published:** 2022-11-09

**Authors:** Liyuan Yang, Yu Zhao, Guozhe Zhang, Linxue Shang, Qun Wang, Sidan Hong, Qingqing Ma, Cuihua Gu

**Affiliations:** 1College of Landscape and Architecture, Zhejiang Agriculture & Forestry University, Hangzhou 311300, China; 2Zhejiang Provincial Key Laboratory of Germplasm Innovation and Utilization for Garden Plants, Zhejiang Agriculture & Forestry University, Hangzhou 311300, China; 3Key Laboratory of National Forestry and Grassland Administration on Germplasm Innovation and Utilization for Southern Garden Plants, Zhejiang Agriculture & Forestry University, Hangzhou 311300, China

**Keywords:** *Heimia myrtifolia*, CAMTA gene family, bioinformatics, tissue specificity, drought stress

## Abstract

Calmodulin-binding transcription factor (CAMTA) is an important component of plant hormone signal transduction, development, and drought resistance. Based on previous transcriptome data, drought resistance genes of the *Heimia myrtifolia* CAMTA transcription factor family were predicted in this study. The physicochemical characteristics of amino acids, subcellular localization, transmembrane structure, GO enrichment, and expression patterns were also examined. The results revealed that *H. myrtifolia* has a total of ten members (*HmCAMTA1~10*). Phylogenetic tree analysis of the HmCAMTA gene family revealed four different branches. The amino acid composition of CAMTA from *H. myrtifolia* and *Punica granatum* was quite similar. In addition, qRT-PCR data showed that the expression levels of *HmCAMTA1*, *HmCAMTA2*, and *HmCAMTA10* genes increased with the deepening of drought, and the peak values appeared in the T4 treatment. Therefore, it is speculated that the above four genes are involved in the response of *H. myrtifolia* to drought stress. Additionally, *HmCAMTA* gene expression was shown to be more abundant in roots and leaves than in other tissues according to tissue-specific expression patterns. This study can be used to learn more about the function of CAMTA family genes and the drought tolerance response mechanism in *H. myrtifolia*.

## 1. Introduction

In recent years, the problem of climate change has become serious [1]. Extreme weather occurs frequently, and drought has attracted attention. Drought is a key abiotic stress factor affecting plant growth and development. It can cause cell dehydration, plasma wall separation, and damage to the cell membrane and enzyme systems, leading to metabolic imbalance and ultimately changing the direction of plant growth [2,3]. Interestingly, plants have evolved a variety of defense systems over the course of their long evolutionary process to fully or partially withstand harmful environmental conditions such as drought [4]. For example, the effects of drought can be mitigated by promoting root growth, increasing root/shoot ratio, destroying photosynthetic pigments, reducing photosynthetic efficiency, and slowing down plant growth [5,6,7]. In addition, at the level of molecular regulation, the expression of a series of drought stress-related genes was induced or inhibited, including genes directly involved in drought stress and transcription factors regulating the expression levels of these genes [8,9,10].

As an important second messenger, Ca^2+^ is sensed by calmodulin and forms a complex that will bind a variety of transcription factors and then regulate downstream target genes and participate in the regulation of the growth and development of organisms [11]. Regulators that are recognized by the Ca^2+^/CaM complex and can regulate downstream target genes are key links in signal transduction pathways [12]. Calmodulin-binding transcription activator (CAMTA) is an important calmodulin-binding transcription factor [13]. As one of the CAM-binding proteins ubiquitous in multicellular eukaryotes, it responds to a series of environmental stresses and hormone signals and plays an important regulatory role in plant biotic stress, abiotic stress, and plant growth and development [14,15]. CAMTA was first discovered in *Nicotiana Tabacum* in 2000 [16]. CAMTA is a family of highly conserved transcription factors in evolution, and it is a multi-gene family. CAMTA consists of CG-1 DNA binding domain, transcription factor immunoglobulin-like DNA-binding domain (TIG) and Ankyrin Repeat (ANK), and the IQ motif of tandem repetition. At time of writing, there are six members of this family in *Arabidopsis thaliana*. These six transcription factors can jointly regulate the expression of genes related to biological and abiotic stress, and plant growth and development [17,18]. It was found that *AtCAMTA1* and *AtCAMTA3* play roles in plant resistance to freezing injury, drought stress, regulation of auxin, and salicylic acid [11,19,20,21,22]. *AtCAMTA6* supports plant resistance to salt stress [23]. *AtCAMTA4* plays a key role in the plant defense response to wheat whitefly and low-temperature stress [24]. CAMTA transcriptional activator families have been identified in a variety of other plants, such as *N. tabacum*, *Populus trichocarpa*, *Phaseolus vulgaris*, *Gossypium*, etc. [25,26,27,28].

*Heimia myrtifolia* is a shrub of *Heimia* in the Lythraceae family. It has golden yellow flowers that bloom in the summer and can be used as a background for garden hedges and flower border [29]. It can also improve the quality of city water. Its potential for use in gardens and high garden value result from this. *H. myrtifolia* is native to Brazil, and later introduced to Zhejiang, Shanghai, Guangxi and other provinces in China. *H. myrtifolia* likes warm, humid, sunny environments and does not tolerate drought. Currently, studies on the classification and application of phenolic compounds, flavonoid components in leaves, abiotic stress resistance mechanisms, chloroplast genome analysis, drought stress and so on have been carried out [30,31,32,33]. According to the studies mentioned above, despite its enormous potential, its growth was mostly constrained by seasonal drought and a lack of moisture. The leaves of *H. myrtifolia* will wilt, even turn yellow, and become deciduous, which will severely restrict the capacity of the plant to grow. Based on previous research, the relevant molecular regulation mechanism of *H. myrtifolia* to resist drought stress has been revealed. In addition, studies in the model plant *A. thaliana* have demonstrated that members of the CAMTA gene family were involved in the response to drought stress and controlled the expression of genes relevant to drought stress. However, studies on CAMTA transcription factors in *H. myrtifolia* have not been reported. In this study, we used bioinformatics techniques based on transcriptome data of *H. myrtifolia* to identify members of the HmCAMTA gene family. Under simulated drought stress, the physical and chemical characteristics, phylogenetic evolution, GO enrichment, expression profile, tissue expression, and gene expression patterns in *H. myrtifolia* leaves were studied. The results provided a reference for further exploring the biological function of the HmCAMTA gene family and the molecular mechanism of their response to drought stress.

## 2. Results

### 2.1. Identification and Physicochemical Properties Analysis of HmCAMTA Gene Family

Program of BLASTP searches were run to search CAMTA proteins in *H. myrtifolia* transcriptome data (the accession number(s) can be found below: https://www.ncbi.nlm.nih.gov/, PRJNA804698) (accessed on 20 February 2022) using AtCAMTA proteins as queries. As a result, 58 unique sequences wee identified. In addition, 10 putative HmCAMTA genes were discovered using an HMMER3 search in the *H. myrtifolia* protein database using the CG-1 DNA binding domain (Pfam03859), IQ motifs (Pfam00612), ANK anchor protein repeats (Pfam12796), and TIG domain involved in non-specific DNA binding (PFam01833). Finally, the Batch Web CD-search Tool and SMART website were used for domain inspection. Duplicate genes without conserved domains were removed. And 10 *HmCAMTA* candidate genes were obtained and named *HmCAMTA1~10*. Further, physical and chemical properties of HmCAMTA were predicted through ExPasy and SOPMA websites (Table 1). The number of amino acids was 864~1080 aa, the molecular weight was 98,275.23~120,654.14 Da, and the instability index was 76.56~79.96. Two proteins with isoelectric points greater than 7 were alkaline, and the other eight were acidic. The average hydrophilicity was negative, and the subcellular localization results showed that all members of the HmCAMTA gene family were located in the nucleus, indicating that all members of the HmCAMTA gene family were intracellular hydrophobic proteins.

### 2.2. Prediction of the Secondary and Tertiary Protein Structure of HmCAMTA Gene Family

The secondary structure of HmCAMTA was analyzed (Table 2): The secondary structure of HmCAMTA (HmCAMTA3, 4, 5, 7, 8, 9, 10) proteins was mainly alpha helix, accounting for 70%, followed by random coil, and then extended strand, with the least beta rotation. The secondary structure of the other 3 HmCAMTA (HmCAMTA1, 2, 6) proteins was mainly alpha helix, accounting for 30%. Next was random crimping, followed by extended strand, and the beta rotation was the least. Figure 1 showed that random curl and alpha helix played a major role in maintaining tertiary structure stability while beta rotation played a major role in modification. The genetic relationships of HmCAMTA1, HmCAMTA2, and HmCAMTA3 were close, and the tertiary structure models of proteins were highly similar. The tertiary protein structures of HmCAMTA4, HmCAMTA5, HmCAMTA6, and HmCAMTA10 were highly similar. And the tertiary protein structures of HmCAMTA7, HmCAMTA8, and HmCAMTA9 were also highly similar.

### 2.3. Analysis of Conserved Domain and Motif of HmCAMTA Gene Family

The conserved HmCAMTA domain was analyzed (Figure 2C). The predicted 10 members of the HmCAMTA gene family all contained the CG-1 DNA binding domain, IQ motif, ANK_2 anchor protein repeats, and TIG domain involved in non-specific DNA binding. Four HmCAMTA proteins contain an ANK sequence. The MEME software was used to analyze the conserved motifs of 10 HmCAMTA proteins, and a total of 10 conserved motifs were identified and named Motif 110 (Figure 3). The results showed that all 10 HmCAMTA proteins contained motifs 1~10. According to the results, the length of motif varied between 21 and 50 conserved amino acids. The longest motif, having 50 conserved amino acids, was from HmCAMTA 1 to 7. The shortest motif, motif 10, had 21 conserved amino acids. The results in Figure 2B indicate that all 10 HmCAMTA proteins included the highly conserved motifs 1~10. Among them, there were two motif 1 in HmCAMTA1 and HmCAMTA2 proteins. Additionally, the majority of the individuals in the closely linked phylogenetic trees showed similar motifs in the same alignment mode and location (Figure 2A).

### 2.4. Multiple Sequence Alignment and Phylogenetic Analysis of HmCAMTA

Jalview was used to perform multiple sequence alignment on 10 HmCAMTA and 6 AtCAMTA proteins. To determine whether and where conserved protein domains exist. Four conserved domains of the CAMTA protein are depicted in Figure 4. In order to understand the evolution of HmCAMTA transcriptional activator family, phylogenetic evolutionary trees were constructed based on CAMTA protein sequences in *A. thaliana*, *Oryza sativa*, *N. tabacum*, *P. granatum*, *Zea mays*, *Solanum lycopersicum*, *P. trichocarpa*, *Micromonas commoda*, *Amborella trichopoda*, *Physcomitrella patens* and *Selaginella moellendorffii*. The number of CAMTA proteins in these species was 6, 7, 19, 6, 7, 7, 18, 1, 4, 2 and 8, respectively, as shown in Figure 5. In addition, protein names, accession numbers, and protein lengths of CAMTA family members of the above species were shown in Appendix A. The 95 CAMTA proteins were divided into 6 branches (Group I to Group VI), and HmCAMTA was distributed into 4 branches (Group I to Group IV). Among them, HmCAMTA1 and HmCAMTA2 were distributed in Group IV HmCAMTA3 was isolated in Group III, followed by HmCAMTA7, HmCAMTA8, and HmCAMTA9 in Group I, and HmCAMTA4, HmCAMTA5, HmCAMTA6, and HmCAMTA10 in Group II. Figure 5 showed that all HmCAMTA proteins clustered with the CAMTA family of *P. granatum*. And both *H. myrtifolia* and *P. granatum* belong to the order Myrtales. Therefore, it was speculated that HmCAMTA was closely related to the CAMTA family in *P. granatum*. Interestingly, Group V consists only of monocotyledonous species, with bryophyte and fern clustered on Group VI. These results suggest that Group V may be a gene specific to monocot plants such as *Z. mays* and *O. sativa*, and Group VI may be a gene specific to bryophytes and ferns.

In addition, some studies have shown that gene family members located in the same subgroup share common origins and conserved functions, and the function of homologous genes can be determined by the function of known genes. For instance, *AtCAMTA1* and *AtCAMTA3* were involved in plant resilience to drought stress and frost damage [20,21]. *AtCAMTA4* was essential for the defense response of plants to low temperature stress, while *AtCAMTA6* enhanced plant tolerance to salt stress [12]. The above studies can be used to infer the related functions of HmCAMTA family genes according to the relative distance of the evolutionary tree.

### 2.5. Annotation and Enrichment Analysis of HmCAMTA Genes

It displayed functional annotations for 10 *HmCAMTA* genes. All genes were involved in cellular components (CCs), biological processes (BPs), and molecular functions (MFs). We examined the GO enrichment data of 10 *HmCAMTA* genes to predict their biological processes. As depicted in Figure 6, various functional categories-including cellular response to stimulus, response to stress, cellular metabolic process, organic substance metabolic process, primary metabolic process, response to abiotic stimulus and nitrogen compound metabolic process are involved in biological processes. It was mostly enriched in the way cellular responses to stimulus and stress are handled. In addition, the genes HmCAMTA4, HmCAMTA5, HmCAMTA6 and HmCAMTA10 in the functions “response to stress”, “cellular response to stimulus” and “response to abiotic stimulus” were not enriched. Interestingly, the genes *HmCAMTA4*, *HmCAMTA5*, *HmCAMTA6* and *HmCAMTA10* were located in Group II of the evolutionary tree. It can be inferred that Group II may not participate in drought stress to a greater or lesser extent.

### 2.6. Analysis of FPKM Values of Differentially Expressed HmCAMTA in Drought Plants

Using the FPKM value in the *H. myrtifolia* transcriptome data (accession number(s): PRJNA804698), the expression level of the HmCAMTA genes following drought treatment was revealed. Among them, the drought treatment groups were S1, S2 and S3, and the soil moisture content was 65~75%, 30~45%, and 5~15%, respectively. The expression profiles of 10 *HmCAMTA* genes following drought treatment were obtained through data extraction and screening (Figure 7). *HmCAMTA2*, *HmCAMTA7*, and *HmCAMTA10* genes were among those most highly expressed during the drought treatment. The expressions of *HmCAMTA2*, *HmCAMTA7 HmCAMTA9* and *HmCAMTA10* genes were up-regulated after drought treatment. After being exposed to drought, the expression of other *HmCAMTA* genes did not change significantly. In general, with the deepening of drought, some members of the HmCAMTA family showed an up-regulated expression trend. The *HmCAMTA2*, *HmCAMTA7*, and *HmCAMTA10* genes may play an important role in the resistance to drought stress, which also needs to be verified by experiments in the later stage. In conclusion, the differential expression patterns of the same family members in different organizations suggest that HmCAMTA members play important functions in different organizations of *H. myrtifolia*.

### 2.7. Tissue Specific Expression Analysis of HmCAMTA Genes

To investigate whether the HmCAMTA gene family plays a role in tissue development, the expression of *HmCAMTA* genes in leaves, flowers, stems and roots was analyzed by qRT-PCR. As can be seen from Figure 8, Except *HmCAMTA6* and *HmCAMTA8* genes, other *HmCAMTA* genes were highly expressed in roots. In addition, the expression levels of *HmCAMTA6*, *HmCAMTA8* and *HmCAMTA10* genes were higher in leaves. Interestingly, the expression of *HmCAMTA10* gene was higher in both roots and leaves. However, the expression of *HmCAMTA* genes in other tissues was not high. The expression of HmCAMTA gene family members was different in different tissues, suggesting that the functions of *HmCAMTA* genes were differentiated.

### 2.8. Expression Patterns of HmCAMTA Genes under Drought Stress

The changes in gene expression during drought stress are depicted in Figure 9 through quantitative analysis of the expression of HmCAMTA family genes. The results showed that except for *HmCAMTA3*, *HmCAMTA4*, *HmCAMTA5*, and *HmCAMTA6* genes, the expression levels of other genes gradually increased with the deepening of drought, and the expression level was the highest under T4 treatment. Among them, the expression of the *HmCAMTA2* gene was the highest of all the genes. Under T4 treatment, the expression of *HmCAMTA2* gene was 1.9~9.8 times that of other genes mentioned above. These results indicated that the *HmCAMTA2* gene might play a role in drought stress. Besides, the expression of *HmCAMTA3* and *HmCAMTA4* genes gradually decreased. The changes in the expression of *HmCAMTA5* and *HmCAMTA6* genes were a process of first decreasing, then increasing, and finally decreasing. And these results were basically consistent with the transcriptome sequencing results. Therefore, the HmCAMTA gene family might be involved in the signal transduction process after drought stress.

## 3. Discussion

Understanding the molecular mechanism of plant response to drought stress is of great significance for analyzing plant life activities and further affecting agricultural production. The CAMTA gene family is crucial in the transduction of the stress signal in plants because they are calmodulin transcriptional activators [14]. It has been confirmed in many plants, but there are few related reports on the *CAMTA* gene of *H. myrtifolia*. The nucleotide and protein sequences of *CAMTA* genes in *A. thaliana* were used as references in this study. The transcriptome of *H. myrtifolia* revealed a total of 10 *HmCAMTA* genes. So far, researchers have identified the transcription factor in seven of *P. trichocarpa*, six to nine in *Gossypium*, six in *S. lycopersicum*, and eight in *P. vulgaris*, with little variation in the number of members of this family [25,27,28,34]. The physicochemical properties showed that there were no significant differences in protein length and relative molecular weight among HmCAMTA family members. The isoelectric points of the expressed proteins varied widely, and the instability coefficient was greater than 40, indicating that the expressed products were not stable. The results of the subcellular prediction revealed that the nucleus was the only location for all of the members. Additionally, recent research has demonstrated that *CAMTA* genes are typically found in the nucleus and has confirmed their primary role as transcription factors that control the expression of other genes [25,28]. The CaMB domain binds CaM in a Ca^2+^-dependent manner, whereas the IQ motif binds CaM in a Ca^2+^-independent manner, according to earlier research [17,35]. All members of the HmCAMTA gene family have CG-1 and IQ structures, suggesting that HmCAMTA can interact with CaM both in a Ca^2+^-dependent and Ca^2+^-independent manner. A phylogenetic tree was created using multi-species association to analyze the phylogenetic relationship and evolutionary link of the CAMTA gene family in plants. Phylogenetic analysis showed that HmCAMTA proteins could be divided into four branches, while other species were generally divided into four or three branches, such as *Linum usitatissimum* (3), *P. trichocarpa* (3), and *Z. mays* (4) [25,36,37]. The distribution of AtCAMTA in other phylogenetic groups was consistent with that in this phylogenetic group, so it was reasonable for HmCAMTA to be divided into four branches. Interestingly, the results of the evolutionary tree show some differences between plant types. Group V consists only of monocotyledonous species, Group Ⅵ consists of mosses and ferns. Other groups had a mixture of plant types. In addition, other studies have shown similar results. Runqing Yue et al. divided the CAMTA evolutionary tree into four subgroups [37]. In Group I a, there were only monocotyledons, while in Group I b, there were no monocotyledons. In Groups II and III, monocotyledons and dicotyledons exist. The cotton CAMTA family was also divided into four groups [28]. Among them, the CAMTA genes from bryophytes and lycophus plants clustered into Group III, while none of the members of the cotton CAMTA gene family belonged to Group III. Six subgroups (Ia, Ib, IVa, IVb, Va, and Vb) of group I, IV, and V were only found in higher terrestrial plants, respectively, whileCAMTA of subgroups I b and IV b was only found in monocotyledons. The CAMTA gene was present in both monocotyledons and dicotyledons except for the three groups. Therefore, it can be speculated that the *HmCAMTA* genes might have shared a common ancestor before the divergence of lower non-fowering (bryophytes, algaes and ferns) and higher fowering plants. Lineage-specific divergence and expansion events occurred in higher plants after splitting from lower plants. In addition, this study showed that the CAMTA of *H. myrtifolia* and *P. granatum* were highly homologous and closely related, which was consistent with their close evolutionary distance, which belonged to the same family of Lythraceae. HmCAMTA1, HmCAMTA2, and AtCAMTA1 were all found on the same branch of the tree. The biological functions of AtCAMTA1 members in this branch have been well studied. AtCAMTA1 is a transcription factor in response to drought stress, which plays an important role in plant drought stress through positive regulation under drought stress. While HmCAMTA1 and HmCAMTA2 were significantly and continuously upregulated under drought stress, it is speculated that HmCAMTA1 and HmCAMTA2 may play a role in *H. myrtifolia* under drought stress through biochemical reactions similar to AtCAMTA1. Interestingly, when the conserved motifs of the *HmCAMTA* genes were analyzed, it was found that there was one more motif 1 in *HmCAMTA1* and *HmCAMTA2* than in other genes, suggesting that it might be an important structure involved in resisting drought stress.

Additionally, GO annotation and enrichment analysis revealed that the HmCAMTA gene family was connected to several stress-related functions, emphasizing the importance of the family in the stress response. It is speculated that this family may contribute to how individuals react to drought stress. Moreover, *AtCAMTA1* gene GO enrichment analysis showed that functions including “reaction to abscisic acid stimulation,” “response to salt stress,” etc. were significantly enriched, which was in line with the results in Figure 7 [20]. In addition, *AtCAMTA1* belongs to the same subgroup as *HmCAMTA1* and *HmCAMTA2* in the evolutionary tree, making it possible that they have functional similarity. In this study, the differential expression of the HmCAMTA gene family in plants under various drought treatments was examined using the FPKM values in the *H. myrtifolia* transcriptome sequencing data. The results demonstrate that *HmCAMTA2*, *HmCAMTA7* and *HmCAMTA10* genes were different degrees of up-regulated in drought-stressed plants compared to normal plants, whereas the expression of the remaining genes was ambiguous. It is speculated that the above three genes may be the key genes to resist drought. But that remains to be proven. Therefore, qRT-PCR was used to detect the expression levels of *HmCAMTA* family genes under different drought degrees to verify the correctness of transcriptome expression profiles. The qRT-PCR verification showed that the expression of *HmCAMTA* genes was consistent with the change trend of the transcript. The correctness of the expression profile was basically verified. According to the results of qRT-PCR, the expression levels of *HmCAMTA1*, *HmCAMTA2* and *HmCAMTA10* genes changed significantly compared with other genes, and their expression levels increased with the deepening of drought. There were differences between the two methods, but it was found that the expression levels of *HmCAMTA2* and *HmCAMTA10* genes were higher under the two methods. It can be speculated that these two genes may be the genes of the HmCAMTA gene family involved in drought stress. In other studies, the *AtCAMTA1* mutant of *A. thaliana* exhibited greater drought sensitivity, revealing its ability to respond to drought [20]. Similarly, the *GmCAMTA12* gene of *Glycine max* improved the drought survival and growth performance of transgenic *A. thaliana* [38]. Therefore, the drought resistance ability of the *HmCAMTA* gene can be further studied by transgenic technology. Different genes are expressed differently in different tissues in response to drought stress [39,40,41]. The expression levels of every member of the HmCAMTA gene family in various tissues were examined in this work. The results showed that different *HmCAMTA* genes were specifically expressed in root and leaf tissues, respectively. Related studies showed that *BnCAMTA* gene was highly expressed in the stem, cotyledon and true leaves of *Brassica napus* [42]. Some CAMTA genes in *S. lycopersicum* showed strong expression in fruit, indicating that their potential role is closely related to fruit development and ripening [34]. Moreover, the spatio-temporal expression pattern showed that the *ZmCAMTA* gene was highly expressed in the roots of most *Z. mays* plants [37], which was similar to the expression pattern of *H. myrtifolia*. However, how the *HmCAMTA* genes play a role in the root system under different environmental stresses needs to be further studied.

In this study, bioinformatics and qRT-PCR were used to identify 10 *HmCAMTA* genes at the transcriptome level. The findings demonstrated that the genes in this family were capable of participating in the response to drought stress in addition to having evident tissue specificity. These findings provide a theoretical framework for more research on the function of the HmCAMTA gene in the growth and development of *H. myrtifolia* as well as its adaptation to drought stress.

## 4. Materials and Methods

### 4.1. Plant Materials

The experimental materials were obtained from two-year old cuttings with similar growth in the greenhouse of Zhejiang A&F University and planted in pots. Five water treatments (CK, T1, T2, T3, T4) were set and watered timely to keep the relative soil water content at 65~75%, 45~60%, 30~45%, 15~30% and 5~15%, respectively. Three replicates were set for each treatment, and samples were taken after 10 days of treatment, respectively. The leaves of *H. myrtifolia* were cut, flash-frozen in liquid nitrogen, and stored at −80 °C for later use.

### 4.2. Identification and Physicochemical Properties Analysis of HmCAMTA Gene Family Members

The Pfam database (http://pfam.xfam.org/) (accessed on 1 March 2022) [43] of the Hidden Markov model file contained the CG-1 DNA binding domain (Pfam03859), IQ motifs (Pfam00612), ANK anchor protein repeats (Pfam12796), and a TIG domain involved in non-specific DNA binding (PFam01833). The HMMER program (Robert, D.F.; Ashburn, VA, USA) was used to model all the protein sequences in the transcriptome of *H. myrtifolia* (the accession number(s) can be found below: https://www.ncbi.nlm.nih.gov/, PRJNA804698) (accessed on 20 February 2022), and all the sequences containing CAMTA conserved domains were obtained. At the same time, from TAIR database (https://www.arabidopsis.org/index.jsp) (accessed on 25 February 2022) [44] to download *A. thaliana* CAMTA protein sequence as the reference sequence, used BLASTp (https://www.ncbi.nlm.nih.gov/) (accessed on 28 March 2022) compare and obtain homologous genes. After removing the redundancy of the gene sets obtained by the two methods. Furthermore, conservative structure domain analysis was performed using the Batch Web CD-search Tool (https://www.ncbi.nlm.nih.gov/Structure/bwrpsb/bwrpsb.cgi) (accessed on 28 March 2022) [45] and Smart (http://smart.embl.de/) (accessed on 28 March 2022) [46]. Finally, the protein sequences containing typical domains of this family were identified as CAMTA family members of *H. myrtifolia*. ExPASy ProtParam (https://web.expasy.org/protparam/) (accessed on 4 March 2022) [47] uses online tools to calculate protein, amino acid, isoelectric point, molecular weight, hydrophobicity, and other basic parameters. For subcellular localization, Plant-mPloc (http://www.csbio.sjtu.edu.cn/bioinf/plant-multi/#) (accessed on 4 May 2022) [48] was used.

### 4.3. Prediction of Secondary and Tertiary Protein Structure of HmCAMTA Gene Family

The secondary structure prediction of HmCAMTA protein sequence was performed by SPOMA Secondary structure prediction (https://npsa-prabi.ibcp.fr/cgi-bin/npsa_automat.pl?page=npsa_sopma.html) (accessed on 5 May 2022) [49]. The tertiary structure of CAMTA gene family protein was predicted by Swiss-Model (https://swissmodel.expasy.org/interactive) (accessed on 5 May 2022) [50].

### 4.4. Analysis of Conserved Domain and Motif of HmCAMTA

The Batch Web CD-Search Tool was used to search the sequential conservative domain, and then TBtools (Chen, C.J.; Guangzhou, China) [51] software was used for graphical display. The conserved motifs of CAMTA family genes were predicted by MEME (https://meme-suite.org/meme/) (accessed on 25 April 2022) online service and visualized by TBtools software.

### 4.5. Multiple Sequence Alignment and Phylogenetic Analysis of HmCAMTA

The amino acid sequences of *H. myrtifolia* and *A. thaliana* were compared using the Jalview software (Andrew, M.W.; Cambridge, MA, USA). Download *O. sativa, N. Tabacum, P. Granatum, Z. Mays, S. lycopersicum* and *P. trichocarpa* from PlantTFdb (http://planttfdb.gao-lab.org/) (accessed on 30 March 2022) of the CAMTA gene family. In addition, the CAMTA family sequences of *M. commoda*, *A. trichopoda*, *P. patens* and *S. moellendorffii* were downloaded from NCBI (https://www.ncbi.nlm.nih.gov/) (accessed on 2 November 2022). The eleven species’ protein sequences from the CAMTA gene family were used to build phylogenetic trees using Clustal W in the MEGA 11.0 [52] software. The neighborhood method was used to construct the phylogenetic tree, bootstrap was set to 1000, and other parameters were default.

### 4.6. Annotation and Enrichment Analysis in GO Databases

The GO function of the *HmCAMTA* gene was annotated using Eggnog-Mapper (http://eggnog-mapper.embl.de/) (accessed on 23 August 2022), an online tool. The results were compiled using the Eggnog-Mapper auxiliary function of TBtools, and the text files from the downstream analysis were output to GO enrichment analysis for enrichment analysis. Finally, the data was plotted and made beautiful using the internet charting program HIPLOT (https://hiplot.com.cn/) (accessed on 26 August 2022) [53].

### 4.7. Analysis of FPKM Values of Differentially Expressed HmCAMTA in Drought Plants

In this study, the transcriptome data following drought treatment were used to extract the *HmCAMTA* gene expression data under the conditions of S1 (65~75%), S2 (30~45%) and S3 (5~15%). The FPKM values of 10 *HmCAMTA* genes in the transcriptome under different drought treatments were extracted, and the FPKM values were converted into log_2_ values. TBtools software was used to draw heatmaps.

### 4.8. Expression Pattern Analysis of HmCAMTA Gene

Roots, stems, leaves, flowers under normal growth and leaves under different drought treatments (CK, T1, T2, T3, T4) were collected. The samples were rapidly frozen in liquid nitrogen and stored at −80 °C. Total RNAs were extracted using the Instructions of FastPure^®^ Plant Total RNA Isolation Kit (Vazyme, Nanjing, China). Next, HiScript^®^ III All-in-one RT SuperMix perfect for qPCR (Vazyme, Nanjing, China) was applied to reverse transcribe first-strand cDNA from the extracted RNA. Primer 5 software was used to design primers, with *HmGAPDH* as an internal reference gene. Primer sequence information was shown in Table 3. qRT-PCR amplifications were carried out with SYBR^®^ Premix Ex TaqTM (TaKaRa, Dalian, China) in 10 μL volumes using ABI 7300 real-time PCR instrument (Applied Biosystems, Foster City, CA, USA). Three replicates were performed for each selected genes. 2^−ΔΔCt^ method [54] and SPSS software (Norman H. Nie, C. Hadlai Hull and Dale H. Bent; Chicago, IL, USA) were used to analyze the Ct values of each sample and analyze the variance, and the relative expression of *HmCAMTA* genes was calculated. The relative expression levels of different tissues were converted into log2 values, and the heatmap was drawn by TBtools for tissue-specific analysis.

## 5. Conclusions

The bioinformatics method was used in *H. myrtifolia* to identify the ten CAMTA gene family members (*HmCAMTA 1*~*10*) there. Prediction methods included the development of phylogenetic trees, physicochemical features, subcellular localization, gene and protein structure, and conserved motifs of the HmCAMTA gene family. The findings demonstrated that HmCAMTA had various regulatory roles in response to drought stress. It was found that different genes were specifically expressed in roots and leaves. Under a simulated drought condition, the expression patterns of *HmCAMTAs* were reported. It was discovered that *HmCAMTA2* and *HmCAMTA10* were all activated by the drought stress signal and responded to various drought stress treatments. This study provides a theoretical framework for further studies on the biological roles of the CAMTA gene family in *H. myrtifolia*.

## Figures and Tables

**Figure 1 plants-11-03031-f001:**
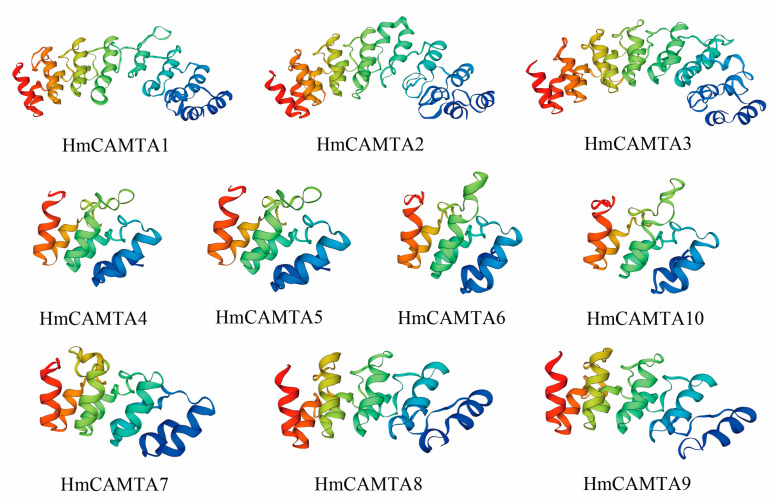
Tertiary structure prediction of HmCAMTA protein.

**Figure 2 plants-11-03031-f002:**
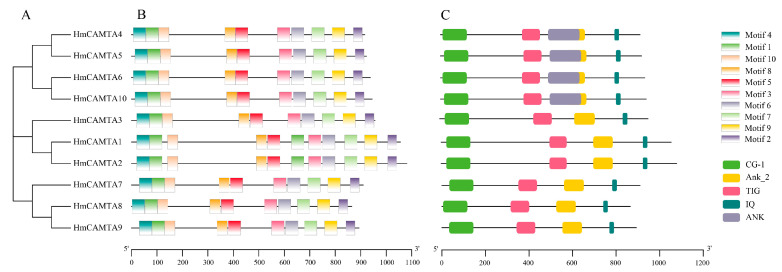
Phylogenetic tree (**A**), protein motif structure (**B**) and domain (**C**) analysis of HmCAMTA gene family. And the modules of different colors represent different motifs or domains.

**Figure 3 plants-11-03031-f003:**
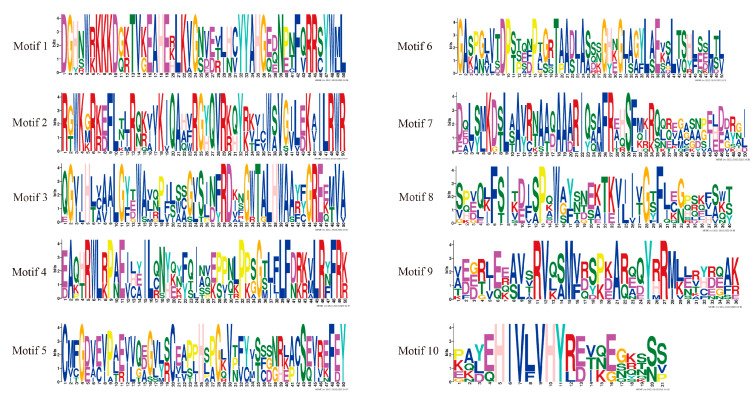
Sequence logo of all identified motifs of HmCAMTA. Motifs 1~10 represent different conservative amino acid sequences, and the fewer amino acids that appear at different positions in each motif, the more conserved it is.

**Figure 4 plants-11-03031-f004:**
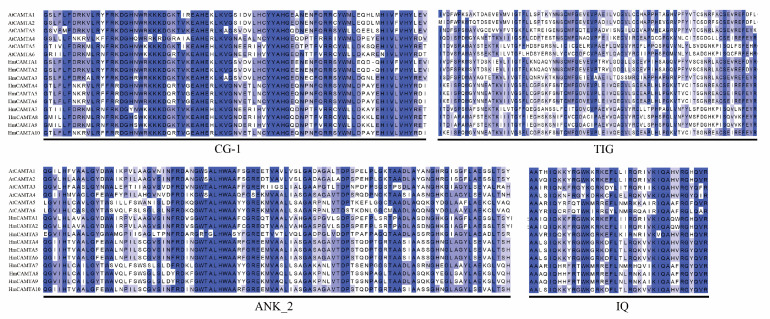
Amino acid sequence alignment of *A. thaliana* and *H. myrtifolia*. The black box lines represent the four domains of HmCAMTA.

**Figure 5 plants-11-03031-f005:**
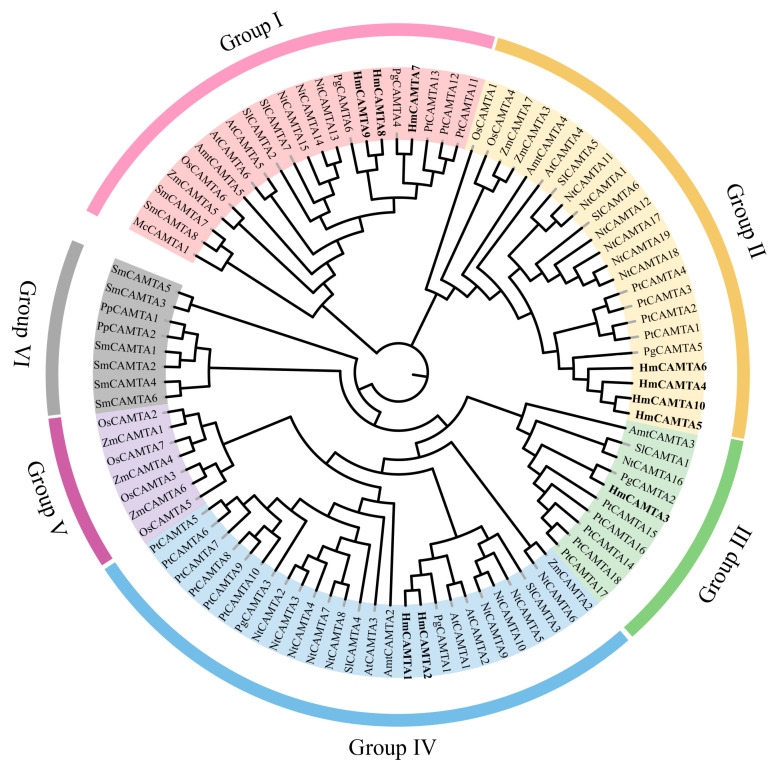
Phylogenetic tree of CAMTA family. Each subgroup is distinguished by a different color. At: *A. thaliana*; Pt: *P. trichocarpa*; Pg: *P. granatum*; Nt: *N. tabacum*; Sl: *S. lycopersicum*; Os: *O. sativa*; Zm: *Z. mays*; Hm: *H. myrtifolia;* Mc: *M. commode*; Amt: *A. trichopoda*; Pp: *P. patens*; Sm: *S. moellendorffii*.

**Figure 6 plants-11-03031-f006:**
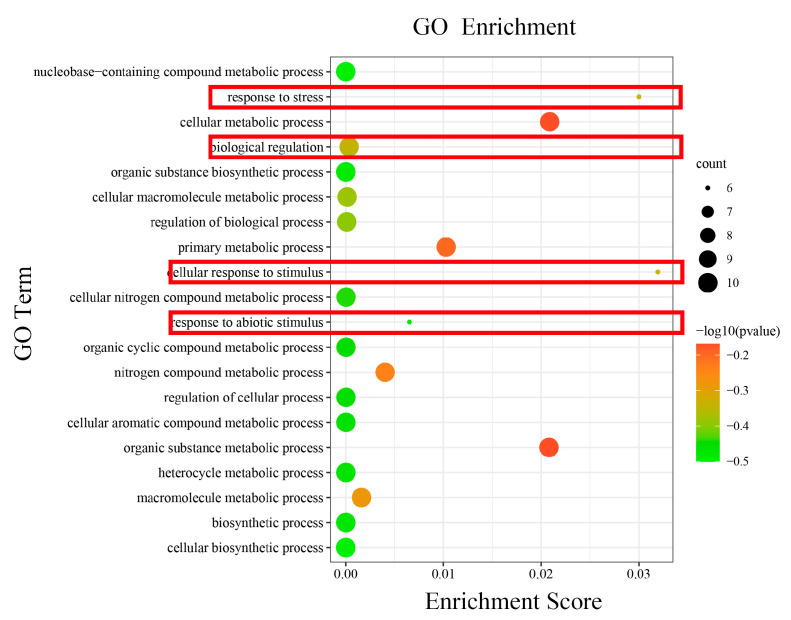
The GO terms enriched of *HmCAMTA* genes. The black circles indicate the number of target genes and different colors indicate the *p*-value. The red boxes represent functions related to drought stress.

**Figure 7 plants-11-03031-f007:**
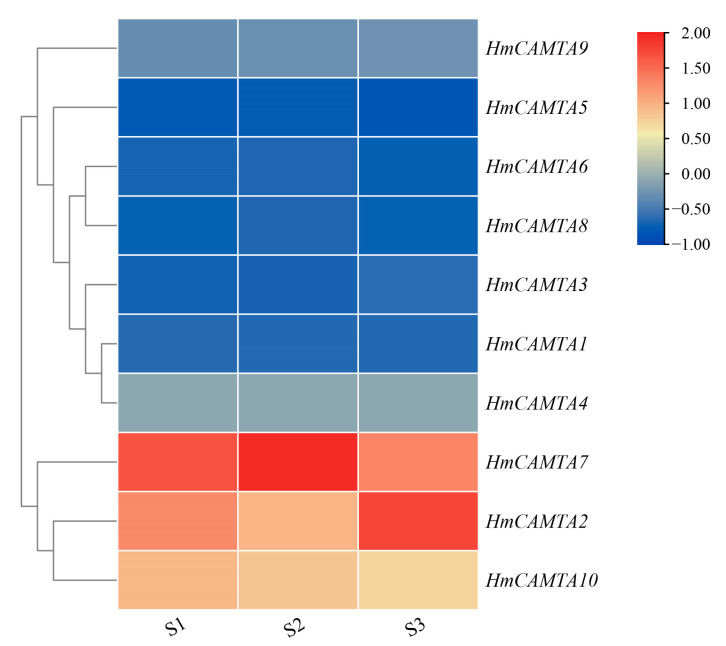
Heatmap expression of *HmCAMTA* genes under drought stress in transcriptome data (S1): Soil water content ranged from 65% to 75%; (S2): Soil water content ranged from 30% to 45%; (S3): Soil water content ranged from 5% to 15%. After homogenization, the value of expression appears in the heatmap, with red representing high expression and blue representing low expression.

**Figure 8 plants-11-03031-f008:**
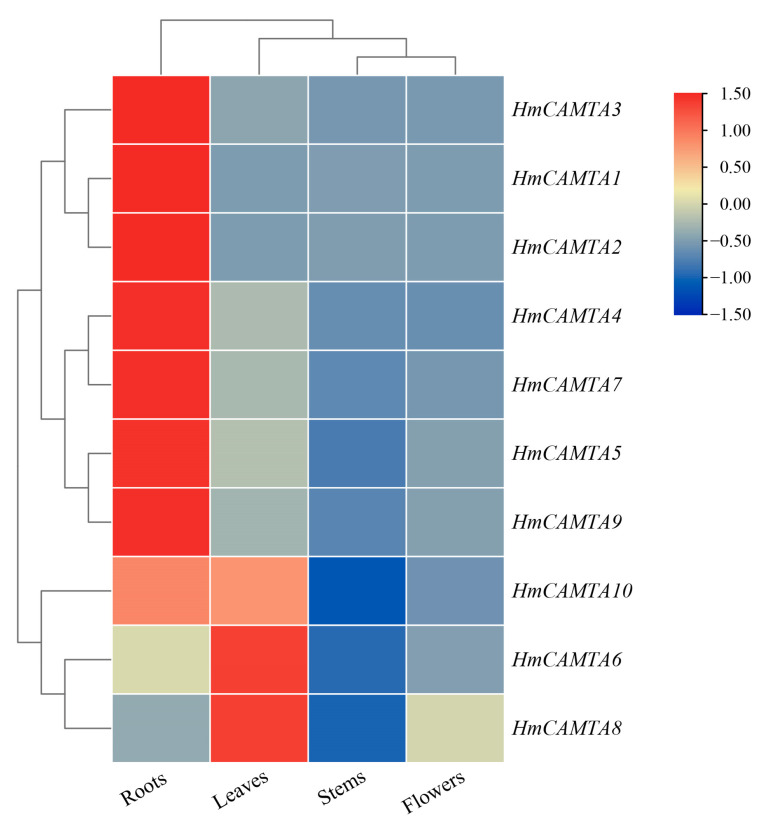
Heatmap analysis of HmCAMTA genes expression in different tissues. After homogenization, the value of expression appears in the heatmap, with red representing high expression and blue representing low expression.

**Figure 9 plants-11-03031-f009:**
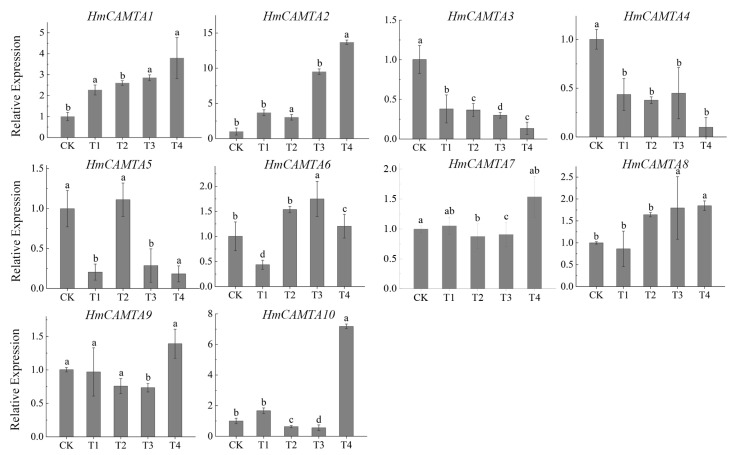
Expression analysis of HmCAMTA gene family under simulated drought stress; significant differences are identified by SPSS with Duncan’s test (*p* < 0.05) and are represented by different letters above the error bars.

**Table 1 plants-11-03031-t001:** Prediction of physicochemical properties of HmCAMTA.

Protein Name	No.of Amino Acid/AA	Molecular Weight/Da	Isoelectric Point	Percentage of the Amino Acids with Highest Content/%	Protein Instability Index	GRAVY	Subcellular Localization
HmCAMTA1	1055	117,935.28	6.14	Ala (A) 6.6	77	−0.538	Nucleus
HmCAMTA2	1080	120,654.14	5.67	Ala (A) 6.6	76.56	−0.541	Nucleus
HmCAMTA3	955	106,484	8.05	Ala (A) 9.3	77.05	−0.467	Nucleus
HmCAMTA4	915	102,392.76	5.83	Ala (A) 6.7	78.64	−0.505	Nucleus
HmCAMTA5	922	103,117.55	5.83	Ala (A) 6.7	78.16	−0.504	Nucleus
HmCAMTA6	937	104,717	5.54	Ala (A) 6.8	77.12	−0.519	Nucleus
HmCAMTA7	909	102,908.94	6.55	Ala (A) 7.7	78.47	−0.504	Nucleus
HmCAMTA8	864	98,275.23	7	Ala (A) 7.6	78.83	−0.5	Nucleus
HmCAMTA9	892	101,172.53	6.61	Ala (A) 7.4	79.96	−0.476	Nucleus
HmCAMTA10	944	105,441.79	5.54	Ala (A) 6.9	76.65	−0.518	Nucleus

**Table 2 plants-11-03031-t002:** Secondary structure prediction of HmCAMTA.

Protein Name	Alpha Helix/%	Beta Turn/%	Random Coil/%	Extended Strand/%
HmCAMTA1	40.85	5.31	45.21	8.63
HmCAMTA2	41.57	5.19	45.19	8.06
HmCAMTA3	44.29	6.07	41.88	7.75
HmCAMTA4	42.51	5.57	41.64	10.27
HmCAMTA5	43.28	5.97	40.46	10.3
HmCAMTA6	41.3	5.55	42.58	10.57
HmCAMTA7	45.43	5.94	39.27	9.35
HmCAMTA8	46.18	6.02	37.73	10.07
HmCAMTA9	45.18	5.72	39.24	9.87
HmCAMTA10	42.27	5.93	41.63	10.17

**Table 3 plants-11-03031-t003:** qRT-PCR primers of *HmCAMTA*.

Gene Name	Forward Primer (5′-3′)	Reverse Primer (5′-3′)
*HmCAMTA1*	AGCCAAACAGCAACCACAGG	ATAAGCCAAGTCCAAACCTAAAGAG
*HmCAMTA2*	AGCCAAACAGCAACCACAGG	ATAAGCCAAGTCCAAACCTAAAGAG
*HmCAMTA3*	TTCTACAGCCCGAGACGAGG	GAATCATCCCAATCCCGACC
*HmCAMTA4*	GAACGAACAACAAGAAAGGCAAGG	GCCGCATAGAAAAGACCCAACT
*HmCAMTA5*	ACGAACAACAAGAAAGGCAAGG	GCCGCATAGAAAAGACCCAACT
*HmCAMTA6*	CAAAGTTGACCTCACGGCACA	AAATCCTTCCGACCCTTCCA
*HmCAMTA7*	CTTCCAAAACCTCGCACCTAAT	AGCTTTTATACCCGCCGATTAC
*HmCAMTA8*	AGTTCCTTAATTTGCGGAATAAGGC	TCCTGTAATCTTCCTGTGCCTTC
*HmCAMTA9*	CTGGGCAGCATACTATGGGAGG	CGGCGAGGTAAGCGGATAAAC
*HmCAMTA10*	CAAAGTTGACCTCACGGCACA	AAATCCTTCCGACCCTTCCA
*HmGAPDH*	AGAAGGTCGTCATTTCTGCCC	TGGTTGTGCAGCTAGCGTTG

## Data Availability

All data in this study can be found in the manuscript or in the Appendix A.

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
