# Peer review of "Identification of CAMTA Gene Family in Heimia myrtifolia and Expression Analysis under Drought Stress"

_plants, 2022, doi:10.3390/plants11223031_

Round 1
Reviewer 1 Report
The manuscript "Identification of CAMTA Gene Family in Heimia myrtifolia 2 and Expression Analysis under Drought Stress" reports 10 CAMTA genes/transcription factors in Heimia myrtifolia. The study would be of interest to researchers in this area and lies in the scope of Plants. Overall, the content is fine and fairly presented. Addressing the following points would somehow improve its quality.
Some other parameters specially promoter analyses must be included as why these transcription factors are drought-reponsive must be justified with more evidence. How these transcription factors sense drought stress and interact with drought-responsive genes?
Similarly, the miRNA prediction may also be included at minimum threshold.
Are the authors doing any functional analyses experiments on HmCAMTA family?
Please name the RNA extraction kit and qPCR machine properly. Do the same for others too.
Why the authors used HmGAPDH as an internal reference instead of others such as Actin/Elongation factor etc?
Several software/online tools used aren't properly cited.
The language needs improvement. A couple of sentences are highlighted for instance.

Reviewer 2 Report
The manuscript by Yang et al. identifies 10 CAMTA genes homologs of H. myrtfolia, and then do an in-silico structural, functional and phylogenetic analysis of the genes. Finally, a spatiotemporal and under drought expression analysis of the genes closes the results section.
The ms needs a lot of corrections and clarifications of techniques before considering the suitability for publication in Plants.
Comments:
Lines 70 – 74: Syntax correction.
Lines 86 – 92: Syntax correction. I can not understand the biological question. “To understand the drought-resistance 86 mechanism of the CAMTA transcription factor family, we used bioinformatics to identify the members of the CAMTA transcription factor family in this study” Especially this sentence needs to be rewritten.
Lines 95 – 99: I do not understand how the analysis and data sorting was performed. “Based on the transcriptome data of H. myrtfolia…” which accession number? In which database? Which Arabidopsis CAMTA gene(s) were used as bait(s)? what are their accession numbers? Then, the authors stated that BLASTP returned 58 results while HMMER3 gave 10, and then 2 more tools were used to check the motifs to conclude in 10 candidate genes. Which were the criteria to eliminate the remaining 48 genes that came back as BLASTP results?
The whole section has to be rewritten and the authors need to provide more detailed information on how the analysis and data mining were performed. They should also provide the accession numbers of the H. myrtfolia (and Arabidopsis) genes identified and analyzed.
Lines 104 – 105: HmCAMTA not HmCAMAT. Please correct the typo.
Lines 146 - 148: I am confused. The authors state in line 61 “at time of writing, there are six members of this family in Arabidopsis thaliana” and here the authors state “Jalview was used to perform multiple sequence alignment on 10 HmCAMTA and 10 AtCAMTA proteins” and then analyze 10 Heimia and 10 Arabidopsis genes. Besides the fact that in Figure 4 they use the AGI codes of the Arabidopsis genes, where the nomenclature AtCAMTA1, 2, etc would be more helpful, how come the 6 Arabidopsis genes became 10? What are the 4 extra genes? Also, the authors show in Figure 2 that 5 conserved domains are identified in HmCAMTA proteins. CG-1, Ank_2, TIG, IQ and ANK. In Figure 4 they discuss that 4 conserved domains (A – D) are identified among Heimia and Arabidopsis, which I assume that they are included among the aforementioned ones. Please clarify if this is the case and then the authors should indicate in Figure 4 which domain is which (CG-1, Ank, etc instead of A, B, C).
Lines 152 – 162: Why did the authors choose to use these species for phylogenetic analysis? A supplementary table with the Species, gene names (eg AtCAMTA1), accession numbers and lengths of the proteins is missing. In the Figure the accession numbers need to be replaced with a more comprehensive nomenclature (HmCAMTA1, AtCAMTA1, etc). As it is now it is hard to follow how the CAMTA members of each species are divided among the phylogenetic groups. Finally in line 162 the authors conclude that Heimia is more closely related to P. granatum. Based on which criteria? The entire analysis and the conclusions drawn from it need to be better explained and discussed by the authors.
Lines 171 – 178 (Figure 6 and GO analysis): The paragraph is too descriptive. This analysis follows a phylogenetic analysis. Is there a correlation between the phylogenetic groups and the GO terms enriched? The way these data are presented seem a little biased towards “stress response”. The stress response terms highlighted by the authors have the smallest gene count (size of the bubble in the chart). I do not argue that the terms are enriched, but how are these results connected to the previous results? Could this analysis show a “functional grouping” of the CAMTA genes based on their phylogeny or maybe previous results from homologous CAMTA genes from other model species?
Line 183 – 185 and Figure 7: “The expression level of the HmCAMTA gene following drought treatment was examined using the transcriptome data of this plant. The expression profiles of 10 HmCAMTA genes following drought treatment were obtained through data extraction and screening” I do not understand and it is not clear in the materials and methods section, did the authors perform transcriptome analysis or did they use publicly available data? In both cases, accession numbers of data deposition are absent and need to be clarified. If the authors performed the analysis they also need to provide sufficient information on the methodology applied for transcriptome analysis (RNA extraction, library construction, platform used, resulted clean reads and mapping %, software and algorithms used for the analysis, etc). Also on Figure 7, what values does the heatmap show? Is it FPKM values, z-scores or fold-changes? The authors do not state this and should also be clarified clearly on the Figure and the text. Also, what do CK, T1 and T2 stand for? It is not explained neither in the figure legend nor the text section describing the results, so it is difficult for me to follow.
Lines 200 – 207: Same as above. What are the values in the heatmap? The authors also state that CAMTA 5, 6 and 8 are highly expressed in leaves. For 6 and 8, there is a marginal peak of expression in leaves as shown in Fig.7, but I can not say the same about CAMTA5. What is the difference between CAMTA5 and CAMTA4 expression in leaves based on Figure 7? Based on the heatmap as it is presented the only obvious conclusion drawn is that CAMTA1 to 5 are predominantly expressed in the root tissues. The authors need to explain the heatmap (what values are drawn) and recheck their analysis and conclusions drawn.
Lines 208 – 221: It is really hard to follow the biology here. First, is the CK, T1 and T2 displayed in Fig. 7 and Fig. 8 the same? Secondly, I see an inconsistency between the results shown in Figs. 7 and 8 concerning drought treatment. Assuming that the conditions displayed in the 2 figures are the same, in Fig 7 CAMTA6 to 10, are all induced upon drought stress, whereas CAMTA1 to 5 are not, or not in the same magnitude. How can the RT-qPCR verification experiments show different results? According to Fig.8 CAMTA1, 2 and 10 seem to be the most induced genes upon drought stress. Also, which normalization method (s) were applied? DCt? DDCt? It needs to be clarified.
The authors need to recheck both their analyses, provide sufficient technical information about the analyses and discuss the differences that are shown between the two experimental approaches.
Reviewer 3 Report
The authors presented very interesting studies aimed at identification and analysis of genes from plant calmodulin-binding transcription activator (CAMTA) family, which play important roles in hormone, signal transduction, developmental regulation, and environmental stress tolerance including drought stress in Heimia myrtifolia. In this species the CAMTA gene family has not been characterized. In the study, the authors identified ten genes of the CAMTA family based on the transcriptome data. The authors present phylogenetic relationship and physicochemical properties of encoded by HmCAMTA genes proteins. In term to predict theirs function the GO terms enrichment analysis was conducted. The expression level of four genes, HmCAMTA1, HmCAMTA2, HmCAMTA8, and HmCAMTA10 were increased under drought stress. The role of identified genes in drought stress response could be explore in further study. The H. myrtifolia is an ornamental plant and also has some
breeding potential and is sesnitive to drought. This knowledge could also be used also in other plant species.
I have few general questions and some remarks.
Is your work based on the same RNA-seq experiment already published by Lin et al., 2022? In the manuscript is only written based on transcriptome data. Please add some more informations.
The Figure 7 - The heat map analysis of the expression of HmCAMTA family genes in different drought treatments is also based on the same RNA-seq experment published by Lin et al., 2022? Under drought only CK, T1 and T2 points are anlysed and in RT-qPCR study also T3 and T4.
It should be clearly written in the manuscript.
Did you perform separate drought stress experiment or this is the same as Lin et al., 2022?
· - The Abstract is well written and reflects objectives and results obtained.
· - The Introduction part is well written. However, I have noticed some repetitions – lines 35 till 37 and the similar 38 till 39.
· - It is not SPOMA websites, but SOPMA. Please correct it. Line 100.
· - Figure 3 is not readable. Please correct it.
· - In Figure 8 , the HmCAMTA gene should be in italic. I have some doubts about results of statistical analysis. For example: HvCAMTA1: CK b, T1 a, T2 b, T3 a; HvCAMTA2: CK b, T1 b, T2a, T3 b; HvCAMTA5: T3 b, T4 a; HvCAMTA7 no statistical indicator et al. Please carefully check all of them. Please add larger description in Material and methods section how analysis was performed. How this relative expression level was counted.
· - Line 372 in Material and methods it is written that material was collected under drought for root, stem and flowers in four points. Is it true?
· - The drought stress treatment description in Material and method needs some additional information: how plants were grown under drought stress treatment, in pots? What is biological repetition in this experiment? One plant and one leaf from this individual plant?
Round 2
Reviewer 1 Report
The authors have addressed all my concerns and the revised manuscript is now ok.
Author Response
Thank you for your comment and thank you very much for your carefulness and responsibility. Your suggestions has played a great role in improving my manuscript. Once again, we thank reviewer 1 for his constructive comments.
Reviewer 2 Report
The revised version of the manuscript is surely improved compared to the first submission. The authors have done most of the revisions requested but there are still some points that I need to point out.
1. Phylogenetic analysis: Group II is solely comprised from monocotyledonous species isoforms (maize and rice) whereas monocots are absent from Groups I and III. Groups IV and V are comprised of both monocot and eudicot isoforms. This is an observation that the authors could analyze more and maybe discuss the phylogenetic divergence of CAMTA family between the two classes. Additionally, the authors could add more species (if available) of CAMTA homologs from lower angiosperms, or gymnosperms, ferns or algae and see where the evolutionary split of this family is. The argument that H. myrtifolia is closer to P. granatum only according to this tree, can not be supported in my opinion.
2. GO analysis: HmCAMTA members are scattered in 4 Groups in the phylogenetic tree. Are there any differences in GO enriched terms among them? What about the Arabidopsis homologs? I made the same argument on the first round of revision.
3. RNA-seq analysis: Sure the idea to split the heatmaps (spatiotemporal and drought stress) is good and improves comprehension. I still don't understand, since the authors do not explain what are the scales of the heatmaps? I can tell for sure that it is not FPKM values. Is it z-scores? log2 fold changes of treatment vs control? This point needs to be definitely explained. Also why -1 value is white? What does this actually mean? Usually white (or black) on heatmaps is 0 value and then there is a scale between green (or blue) for negative values and red for positive. The authors still need to redraw these figures and explain how they did the analysis and what values they did use.
Round 3
Reviewer 2 Report
The authors have made some text additions concerning the phylogeny of HmCAMTA family and also GO terms enrichment and they have also made the changes suggested in the heatmaps of RNA-seq data.
However, I think that the manuscript the way it is, is too descriptive to be considered for publication in Plants.
A more thorough analysis starting from the phylogeny (the authors in their response stated that "We believe it is important to add a few species of lower angiosperms, gymnosperms, ferns or algae based on your recommendation. But, after searching through databases like PlantTFDB, NCBI, and others, the CAMTA family has not been found in lower angiosperms, gymnosperms, ferns or algae". A quick BLAST search in ensembl Plants with AtCAMTA1 as bait returned hits for Chlamydomonas (algae), Amborella trichopoda (basal angiosperm) and Selaginella moellendorffii (lycophyte)).
I think that the authors should search it a little deeper and unravel the diversification of CAMTA family among the plant kingdom. Secondly, the functional classification (GO enrichment, motifs, etc) and gene expression could be separated for each group of HmCAMTA family (as phylogenetic analysis shows) to see differences and describe the gene paralogs, orthologs, etc. This way it would be much more interesting to the reader.
